# In-Season Microcycle Quantification of Professional Women Soccer Players—External, Internal and Wellness Measures

**DOI:** 10.3390/healthcare10040695

**Published:** 2022-04-07

**Authors:** Renato Fernandes, Halil İbrahim Ceylan, Filipe Manuel Clemente, João Paulo Brito, Alexandre Duarte Martins, Hadi Nobari, Victor Machado Reis, Rafael Oliveira

**Affiliations:** 1Sports Science School of Rio Maior—Polytechnic Institute of Santarém, 2040-413 Rio Maior, Portugal; jbrito@esdrm.ipsantarem.pt (J.P.B.); alexandremartins@esdrm.ipsantarem.pt (A.D.M.); 2Life Quality Research Centre, 2040-413 Rio Maior, Portugal; 3Sport Sciences Department, University of Trás-os-Montes e Alto Douro, 5001-801 Vila Real, Portugal; victormachadoreis@gmail.com; 4Physical Education and Sports Teaching Department, Faculty of Kazim Karabekir Education, Ataturk University, Erzurum 25240, Turkey; halil.ibrahimceylan60@gmail.com; 5Escola Superior Desporto e Lazer, Instituto Politécnico de Viana do Castelo, Rua Escola Industrial e Comercial de Nun’Álvares, 4900-347 Viana do Castelo, Portugal; filipe.clemente5@gmail.com; 6Instituto de Telecomunicações, Delegação da Covilhã, 1049-001 Lisboa, Portugal; 7Research Center in Sports Performance, Recreation, Innovation and Technology (SPRINT), 4960-320 Melgaço, Portugal; 8Research Centre in Sport Sciences, Health Sciences and Human Development, 5001-801 Vila Real, Portugal; 9Comprehensive Health Research Centre (CHRC), Departamento de Desporto e Saúde, Escola de Saúde e Desenvolvimento Humano, Universidade de Évora, Largo dos Colegiais, 7004-516 Évora, Portugal; 10HEME Research Group, Faculty of Sport Sciences, University of Extremadura, 10003 Cáceres, Spain; hadi.nobari1@gmail.com; 11Department of Exercise Physiology, Faculty of Educational Sciences and Psychology, University of Mohaghegh Ardabili, 56199-11367 Ardabil, Iran; 12Sports Scientist, Sepahan Football Club, 81887-78473 Isfahan, Iran

**Keywords:** external load, female, football, Hooper Index, internal load, load, match, training load

## Abstract

Although data currently exists pertaining to the intensity in the women’s football match, the knowledge about training is still scarce. Therefore, the aim of this study was to quantify external (locomotor activity) and internal (psychophysiological) intensities, as well as the wellness profile of the typical microcycle from professional female soccer players during the 2019/20 in-season. Ten players (24.6 ± 2.3 years) from an elite Portuguese women soccer team participated in this study. All variables were collected in 87 training session and 15 matches for analysis from the 2019–2020 in-season. Global positioning variables such total distance, high-speed running, acceleration, deceleration and player load were recorded as intensity while Rated Perceived Exertion (RPE) and session-RPE were recorded as internal measures. The Hooper Index (HI) was collected as a wellness parameter. The results showed that internal and external intensity measures were greater in matches compared to trainings during the week (match day minus [MD-], MD-5, MD-4, MD-2), *p* < 0.05 with very large effect size (ES). In the same line, higher internal and external intensity values were found in the beginning of the week while the lowest values were found in MD-2 (*p* < 0.05, with very large ES). Regarding wellness, there was no significant differences in the HI parameters between the training days and match days (*p* > 0.05). This study confirmed the highest intensity values during MD and the lowest on the training session before the MD (MD-2). Moreover, higher training intensities were found in the beginning of the training week sessions which were then reduced when the MD came close. Wellness parameters showed no variation when compared to intensity measures. This study confirmed the hypothesis regarding internal and external intensity but not regarding wellness.

## 1. Introduction

The number of women’s soccer participants is growing [1,2,3] which contributes to the progressive raising of publications related to the topic (in the PubMed, a simple search by “women” AND “soccer” revealed an increment of publications from 75 in 2019 to 125 in 2021). This reveals the concerns of community science to offer more evidence to practical community. Currently, it is recognized that women soccer players covers about 10 to 11 km (km) per professional match [4]. From those km, high-intensity running (18–25 km/h) demands can vary between 718 [4] and 3000 m [5]. These range values are dependent from competitive level, playing position, age-group or event moment of the season and physical fitness [6,7].

Since soccer demands in women soccer are intermittent, this represents a mixed contribution of energetic systems to support the intensities presented in match [7]. Although a greater contribution of low-to-moderate activities, which is consistent with greater aerobic participation, anaerobic systems are also part of the process considering that blood lactate concentrations can vary between 2 and 7 mmol/L [8], and the average heart rate responses in adult women soccer players is around 87% of maximum heart rate [6].

In addition to a well-known knowledge regarding match demands, it is still needed some research to characterize the training process [2]. As an example, a study conducted in elite women soccer players [9] revealed that in the first part of the season a heavy intensity week (week 16) may present 357 arbitrary units (AU) in average per session, while a low intensity week (week 13) may be about 210 AU (values only included weeks with four training sessions, while matches were not included) (considering the multiplication of time of sessions in minutes by the score in a 10-point rate of perceived exertion scale). In the same study, it was found that in a heavy week women player may cover 5090 m per session, while in a low week the average drops to 3870 m [9].

Possibly, these week variations can be affected by the intensities imposed between days of the microcycle. As an example, a study conducted with senior international women players in which was found intra-week variations, in particular greater values of physical and physiological demands in match day (MD)-5 (five days before next match), followed by a progressive intensity decrease until MD-1 [10]. Considering the locomotor activity demands, these intra-week variations can also influence some wellness outcomes. In this sense, a systematic review [11] revealed that the level of a negative correlation between wellness measures (e.g., sleep quality, mood, fatigue, delayed onset muscle soreness, stress) and physiological/physical and locomotor activity demands was small-to-moderate (in general, when there were higher training intensity, the wellness decreased). Even so, the previous systematic review highlighted that not all research was able to find such relationships due to different study designs, methodologies and statistical analysis approaches [11]. Nonetheless, it can be expected to observe intra-week variations of training intensity as well as wellness.

Although there are some cases of studies characterising soccer intra-week variations regarding physiological and locomotor activity demands, and wellness in women soccer, is still need greater research which helps to characterize how the periodization occurs. This type of evidence (intra-week variation) is well-established for men in the last decade where it was observed a tapering on the physiological and locomotor activity demands in last two days before match [12,13,14]. In the case of women, more research is needed to provide possibilities for comparisons between scenarios and contexts. This may help to characterise the reality of training process in professional women soccer players.

Thus, the purpose of this study was to quantify external (locomotor activity) and internal (psychophysiological) intensities, as well as the wellness profile of the typical microcycle from professional women soccer players during the 2019/20 in-season. For this purpose, the match day minus (MD-) approach used in previous studies was applied for data analysis [14,15,16]. It was hypothesized that the training session intensity and Hooper Index values are lower on the training day closer to the next match and that match-day presents the highest intensity of the week.

## 2. Materials and Methods

### 2.1. Design

The observational period occurred during seven months, from September to March (early-to-mid-season) due to the COVID-19 pandemic, which provoked the disruption of training sessions and matches and the suspension of the season in March. Thus, the observational cohort study contemplated 87 training session and 15 matches for analysis from the 2019–2020 in-season.

The players belong to a team that participated in the BPI League, the women’s first League in Portugal. A typical microcycle had three training session and one match per week. For better clarity, the training session occurred in MD-5, MD-4 and MD-2. During MD-3, MD-1 and in the day after the match, the athletes rested.

### 2.2. Participants

Similar to previous studies with small sample sizes [17,18,19], 10 elite women soccer players with a professional experience of 4.9 ± 2.1 years, an age of 24.6 ± 2.3 years, a height of 165 ± 6.0 cm (Seca 220, Hamburg, Germany), a body mass of 58.5 ± 9.3 kg (Seca 220, Hamburg, Germany), and body mass index of 22.3 ± 3.8 kg/m^2^, participated in this study. Moreover, the power of the sample size was estimated through G-Power [20]. The analysis featured 99.2% of actual power, with a total of 10 participants with a *p* < 0.05 and effect-size for 0.6.

We adopted inclusion/exclusion criteria from our previous studies [14,16,21,22] where participants need to achieve a minimum of 80% of the training sessions and an average of 75 min from all matches, while the exclusion criteria were based on becoming injured, ill, sick for two consecutive weeks. Only defenders (*n* = 3), midfielders (*n* = 4), and strikers (*n* = 3) were included for analysis while goalkeepers were removed.

Before the beginning of the study, all explanations about the study design were provided to players. Then, a written informed consent was recorded from all participants. In addition, the study was approved by the research Ethics Committee of the Polytechnic Institute of Santarém, Santarém, Portugal (252020 Desporto) and it was developed according to the requirements of the Declaration of Helsinki.

To control more outside factors that could influence the intensity and wellness over the period of analysis, all participants were asked to maintain their normal diet throughout the study period. To confirm their habits, nutritional intake questionnaire was used to record a 24 h diet over seven days of the week. This procedure was applied in the first and last week of the analysed period following the procedures of our previous study [21]. The size of the food portions, supplements, and other aspects pertaining to an accurate recording of their energy intake were addressed and reviewed for macronutrient composition and total energy intake [23].

### 2.3. Internal Intensity Quantification

The CR10-point scale, adapted by Foster et al., was used 30 min after the end of each training/match session [24]. To avoid non-valid values, all players were previously familiarized with the scale and all answers were provided on google forms, through a tablet. Then, each training/match session value was multiplied by the, respectively, session duration to produce the s-RPE [24,25].

### 2.4. Wellness Quantification

The Hooper Index (HI, 1–7 scale) questionnaire was also used 30 min before each training/match session. The same procedure described in the previous point was used for data collection. This questionnaire has four questions: fatigue, stress, delayed onset muscle soreness (DOMS) (in which 1 is very, very low and 7 is very, very high), and the quality of sleep of the night that preceded the evaluation (in which 1 is very, very bad and 7 is very, very good). Moreover, to produce a final score of HI, data from all questions was summed [26].

### 2.5. External Intensity Quantification

A portable 10 Hz GPS device was used to collect external data (PlayerTek, Catapult Innovations, Melbourne, Australia), which also incorporates a tri-axial 100 Hz accelerometer. These types of GPS devices seem to be the most valid and reliable to use in team sports [27].

Ten minutes before each training session and match, PlayerTek devices were turned on and the players were asked to use them. The devices were turned on and placed in a specific customized vest pocket located on the posterior side of the upper torso fitted tightly to the body, as is typically used in matches. The devices were placed and checked always by the same coach of the team, and the players always used the same device [28].

The measures used for analysis were total distance, high-speed running distance (≥ 15 km/h) [29], number of accelerations (ACC, >1–2 m.s^−2^ [ACC1]; >2–3 m.s^−2^ [ACC2]; >3–4 m.s^−2^ [ACC3]; >4 m.s^−2^ [ACC4]) and decelerations (DEC, <−1–2 m.s^−2^ [DEC1]; <−2–3 m.s^−2^ [DEC2]; <−3–4 m.s^−2^ [DEC3]; <−4 m.s^−2^ [DEC4]) maximal speed, average speed and player load.

### 2.6. Statistical Analysis

Descriptive statistics (mean ± standard deviation, SD) were performed for all measures. All the variables were checked for normality and homoscedasticity, respectively, using the Shapiro–Wilk and Levene tests. Then, repeated measures ANOVA with the Bonferroni post hoc test was calculated to compare the training and match sessions. The *p*-value ≤ 0.05 was used as significant and all the data were analysed using SPSS version 22.0 (SPSS Inc., Chicago, IL, USA) for the Windows statistical software package.

Finally, the Hedges effect-size (ES) was performed to determine the effect magnitude through the difference of two means divided by the standard deviation from the data and the following criteria were used: <0.2 = trivial, 0.2 to 0.6 = small effect, 0.6 to 1.2 = moderate effect, 1.2 to 2.0 = large effect, and >2.0 = very large [30].

## 3. Results

Table 1 shows the MD- differences for duration and running distance variables between training and match days. With the exception for duration, where MD-4 displayed the highest value, all running-based variables showed to be significantly higher in MD with very large effect sizes.

Table 2 shows the MD- differences for accelerometery-based variables, namely, ACC, DEC and player load between training and match days. With the exceptions of player load in MD-5 and ACC4, all variables showed to be significantly higher in MD with very large effect sizes.

Table 3 shows the MD- differences for internal intensity and wellness profile. While variables from HI did not show any significant difference, both RPE and s-RPE showed to be significantly higher in MD with very large effect sizes. There was an exception regarding MD-5 versus MD for s-RPE that showed a large effect size instead of a very large.

## 4. Discussion

The purpose of this study was to quantify external and internal intensities, as well as the wellness profile of a typical microcycle in professional women soccer players during the 2019/20 in-season. The present study indicated that internal and external intensity measures were greater in MD than in weekday training sessions (MD-5, MD-4, MD-2). Moreover, it was found that the wellness parameters (Hooper Index, HI) of the players did not differ significantly between training sessions and matches. During the week, it was observed that the internal and external intensity gradually decreased close to the match, and even in the last two days before the match, there was a serious decrease in the intensity measures so that the players could prepare for the match. Our results supported one part of the hypothesis of the present study that confirmed that the training session intensity values are lower on the training day closer to the next match and that match-day presents the highest intensity of the week. On the other hand, our results did not support that Hooper Index values are lower on the training day closer to the next match.

The present study revealed there was no significant difference in the HI parameters between the three training days and match days in professional women soccer players. Consistent with our results, recent studies reported no intra-week variations (between match day and training sessions) in wellness variables in elite soccer players [13,14]. In our study, it was seen that the wellness measures vary between 2 and 3 on average throughout the microcycle in women soccer players. This result was in agreement with the study carried out by Fernandes et al. [16]. Similarly, Clemente et al. [31] stated that basketball players showed similar health profiles in both training and matches during normal and congested weeks, and the health profiles of the players (very low DOMS, fatigue, and stress and very good sleep quality, around 2 on average) were quite well stated. This shows that both training sessions and matches have similar effects on players and do not sufficiently trigger stress factors. This result is also supported by Clemente et al. [31].

Concerning the RPE and s-RPE, our study demonstrated that MD-5 presented highest intensity than MD-4 (session with highest training duration). Both MD-5 and MD-4 presented higher values than MD-2 (lowest intensity), which revealed a pyramid weekly microcycle shape regarding internal intensity (lower intensity after and before in the training sessions close to the match and higher intensity in the middle week training sessions). Doyle et al. [10] found that the total training intensity in elite female football players increased from MD-7 to MD-5 compared to other training sessions and peaked at MD-5.

In the first and second training sessions of the week (MD-5 and MD-4), the players were exposed to the greatest psychophysiological intensity, which was supported by the study by Ispirlidis et al. [32] and that by Fernandes et al. [16], who noted that fatigue, DOMS and total HI were significantly higher in MD-4 (second weekly training day) as compared with other training and match days in soccer players. As in our study, Romero-Moraleda et al. [29] also reported that training sessions in the middle of the week (MD-4 and MD-3) demonstrated greater intensities in professional female soccer players. Similarly, Clemente et al. [31] stated that fatigue and stress were high due to the training intensity in MD-4 (1st training day of the week) during normal and congested weeks, and accordingly, sleep quality was quite low.

The present study indicated that the intensity of the training (RPE and s-RPE) gradually decreased as the match day approached. Our results were in agreement with Doyle et al. [10], who reported that the total training volume and intensity on both MD-3 and MD-1 decreased during the training sessions close to the match. Similarly, Romero-Moraleda et al. [29] stated that lower intensities implemented in training sessions (MD plus 1, [MD + 1] and MD-2) were closest to MD. Additionally, Malone et al. [15] noted that the training intensity decreased in MD-1 compared to MD-2 and MD-5. In contrast, Oliveira et al. [14] remarked that s-RPE value decreased in an imperfect order from MD-5 to MD-1 in elite male soccer players. In our study, HI scores did not show any change in the days before the match day, which means that during the week, whereas the decrease in RPE and s-RPE from MD-5 as the match day approaches, HI presented no changes. Previous study of Haddad et al. [33] was in line with the present findings. Haddad et al. [33] reported that HI parameters were not significant determinants of perceived effort during traditional soccer training without excessive training intensity. On the other hand, Clemente et al. [34] observed that the relationship between s-RPE and HI was significant and negative (small-medium) in the weeks when there were two official matches and there was no relationship between s-RPE and HI in the weeks when there was one match, as in our study.

Regarding the total distance covered and average speed, the total distance and average speed gradually decreased as the match day approached and this variable reached the highest values on the match day when compared with the training sessions. Consistent with our results, Trewin et al. [35] reported that the female soccer players exhibited lower values for the total distance covered (approx. 4000–5000 m) before the match compared to the other training sessions. Similarly, recent study notified that the total distance covered by female soccer players in the MD-2 (3024 ± 1220 m) was lower than in the MD-4 (4831 ± 860 m) and MD-3 (4975 ± 1318 m) in an one-week training program [29]. Additionally, Doyle et al. [10] noted that MD-5 and MD-2 were the most intense training sessions compared to other training sessions in terms of total distance covered (5933.5 and 5151.5 m, respectively), very high speed running (387.5 and 201 m, respectively) and sprint distance (187.5 and 49 m, respectively) in elite female soccer players. Our results were supported by some studies conducted on top elite male soccer players which indicated that the total distance covered and average speed before the match decreased in MD-1 (in our study, MD-2) compared to other training days (MD-5 and MD-4) [12,14,15,36].

The gradual decrease in internal and external intensity towards the match in the weekly program may be due to the content of the weekday training sessions. Indeed, Romero-Moraleda et al. [29] showed that MD-2 included skill and strategy exercises applied to reduce the training intensity before match day. The present study also used speed and ACC exercises in the execution of skills and goal achievement during MD-2.

Furthermore, the present study demonstrated that the accelerometer-based variables, namely ACC, DEC and player load (except ACC4), decreased from MD-5 to MD-2. In addition, the fact that these variables were higher in MD-5 can be explained by the intense application of small-sided games to players during this training session, and previous studies notified that ACC, DEC and player load were affected by pitch area, and small-sided games increased these variables more [9,37].

Nevertheless, our results for ACC are similar to the previous study that indicated that the number of ACC performed in MD-2 was significantly lower than in MD-5, and there was no difference in DEC between MD-2 and MD-5 in elite female soccer players [10]. Other studies showed that total number of ACC and DEC decreased from MD-4 to MD-2 in elite female soccer players [29], and also that these variables decreased from MD-4 to MD-1 in elite male soccer players [38]. A study of elite male soccer players found lower ACCs values before matches compared to other days [14]. Considering the above studies, Harper et al. [39] stated that greater volume of explosive eccentric actions such as ACCs and DECs during the match could cause greater perceived exertion, muscle damage, neuromuscular fatigue and, as a result, a higher risk of injury. However, the available evidence from the recent study emphasized that the volume of very high intensity running and sprint distance on MD-2 was decreased by 48% and 73%, respectively, showing us how the training session content was modified to optimize player preparation for competition in MD [10]. Additionally, the same authors asserted that the optimum taper period process for international women’s matches are not clear enough in the literature and therefore the most effective periodization methods should be investigated to ensure optimal preparation for matchday performance [10].

In our study, it was seen that the volume and intensity were high on the 1st and 2nd (MD-5, MD-4) training days, and then the volume and intensity decreased towards MD-2. This shows that the tapering strategy (significant reduction in volume and intensity before the match) was used for the weekly training design such as in previous studies [12,35,36,40]. The tapering strategy seems to be a preferred method for reducing the residual fatigue accumulation in the last two days before the match, to optimize the performance and to prepare the players for the match [29]. In addition, previous studies found that the use of the tapering strategy throughout a microcycle created certain performance advantages on players. For instance, in one study, it was reported that reducing the training intensity of professional soccer players by ~25% during taper weeks during matches resulted in a 15% increase in intense and high-intensity activities during matches [41]. In addition, previous studies showed that a reduction in intensity, in the last training session before a match leads to improvements in wellness variables (total HI, DOMS, fatigue, and sleep quality) on a matchday and prevent overtraining in female soccer players [16], and professional basketball players [31]

Soccer matches show the highest intensity in the weekly schedule [14]. The results from our study showed that when comparing MD with the training sessions, the highest RPE and s-RPE values were recorded in MD. This finding was compatible with the study conducted by Fernandes et al. [16] which showed the highest matchday RPE values in female soccer players. Subsequently, the same researchers stated that the match offered lower s-RPE values and interestingly there was no significant difference between MD-5, MD-4 and MD. This result is not in line with the results of present study. In another study, Romero-Moraleda et al. [29] observed that the internal intensity in professional female soccer players was higher in official matches (RPE: 8.4 AU, and s-RPE: 792 AU) compared to training sessions (RPE: 3.1–6.2 AU, and s-RPE: 167–579 AU).

Moreover, the present study revealed that HSR and maximal speed were found to be significantly higher in MD (HSR: 879.7 ± 102.2 m, maximal speed. 26.7 ± 1.2 km/h) compared to weekday training days (HSR: 306.5 ± 33.9–312.8 ± 38.2 m, maximal speed: 22.9 ± 0.5–23.5 ± 0.7 km/h). In addition, ACC and DEC were significantly higher in MD than weekday training days. Our results were supported by the study performed by Romero-Moraleda et al. [29].

Considering the results of the above studies, it was observed that women soccer players are exposed to higher internal and external intensities in matches compared to trainings during the week, intensely engage in high-intensity activities during the match and reach higher maximum speeds in activities. However, Teixeira et al. [42] reported that intensity differences between training and matches may be affected by contextual factors such as the type of weekly schedule, player’s starting status, playing positions, age group, training mode, opponent’s level, the location of matches.

The present study includes some limitations. Firstly, it was carried out with 10 professional adult women soccer players. Clemente et al. [12] found that intensity and tapering methods differed between teams in different countries, so it can be difficult to generalize current results to male or female amateur players of different age categories in different countries. At the same time, the sample size in the present study does not permit the generalization of the results for other teams. Furthermore, players from different positions are exposed to different internal and external intensities [15,29]. The neglect of player positions due to the sample size in our study could be considered the second limitation.

The weekly training design used in this study is valid for women’s soccer teams using the same design. The present study showed that soccer matches generate very high external, internal and wellness measures on the players. Therefore, it is recommended to include different time-efficient training methods (high-speed straight runs, running involving directional changes, repeated short-medium-large sprint ability, time spent in game-based situations, with modifications of pitch dimensions) in training planning, taking into account the player positions in order to cope with the differences in both internal and external intensities between training and official matches [29]. In the literature, the number of studies on women soccer players related to weekly intensity design is still reduced compared to male soccer players. Therefore, the methodology of this study can be comprehensively replicated in women soccer players of different age categories in different countries, with the addition of evaluation of various biochemical parameters, especially after match day.

Increasing studies on women soccer players are important for the development and periodization of training programs. For instance, this study did not address the menstrual cycle. It has been revealed that there are hormonal variations during this cycle and those can affect cardiorespiratory [43] and neuromuscular performance [44], although results are not always consensual [45] and more research is needed. Such analysis would be relevant for future studies to analyse if there are variations of external and internal intensities with respect to the menstrual cycle.

Finally, we suggest that future studies could analyse if the wellness values of the day before influence the training intensity of the day after and if the training intensity of the day influence the HI values of the day after as previously conducted in the Silva et al. study [46].

From a practical point of view, the results of this study can express that one of the main focuses of technical teams is the balanced allocation of the microcycle intensity in order to induce the best performance in the players and simultaneously ensure that the effort management allows to avoid possible harmful effects that may arise from it. This fact was evident in two aspects: (1) in the wellness profiles evidenced by the players with low levels of DOMS, fatigue and stress, plus good sleep quality throughout the microcycle; and (2) the good management of intensity that revealed greater levels in the first and second sessions of the microcycle and then, a reduction until the match. Nowadays, professional technical teams have human resources, whether they are assistant coaches, physical trainers, physiologists, psychologists, or physiotherapists who, working as a team also have the mission of protecting their players and putting them at the highest physical and physiopsychological levels for the competition.

## 5. Conclusions

The present study revealed that the internal (RPE, s-RPE) and external measures (total distance, average speed, maximal speed, high-speed running distance, almost ACC and DEC) were higher on the match day than during the weekday training sessions (MD-5, MD-4, MD-2) throughout the in-season in elite female soccer players. In addition, it was observed that HI values did not differ significantly between training sessions and matches during the week. Finally, the present study showed that players generally reached the highest internal and external intensity in MD-5 and after that, intensity decreased until the MD.

This study confirmed the hypothesis regarding internal and external intensity but not regarding wellness. It is recommended to consider the results of the current study in the weekly intensity distribution in female soccer players and accordingly in the optimal adjustment of the relationship between intensity and rest.

## Figures and Tables

**Table 1 healthcare-10-00695-t001:** External Intensity by running-based variables during training and matches for squad average (mean ± SD).

Day	Duration (min)	Total Distance (m)	Average Speed (m/min)	Maximal Speed (km/h)	HSR (m)
MD-5	85.1 ± 2.8	5121.6 ± 82.2 ^a,b,c,^*	60.6 ± 1.7 ^a,b,c,^*	23.3 ± 0.5 ^c,^*	306.5 ± 33.9 ^c,^*
MD-4	90.9 ± 2.5 ^b,^*	4638.6 ± 73.0 ^b,c,^*	53.5 ± 1.8 ^b,c,^*	23.5 ± 0.7 ^c,^*	312.8 ± 38.2 ^c,^*
MD-2	78.3 ± 2.1	3857.5 ± 73.0 ^c,^*	47.8 ± 1.2 ^c,^*	22.9 ± 0.5 ^c,^*	311.3 ± 22.2 ^c,^*
MD	87.2 ± 2.0	7616.1 ± 395.2	89.9 ± 5.4	26.7 ± 1.2	879.7 ± 102.2

MD, match-day; MD-, matchday minus (5, 4, 2); min, minutes; m, meters; HSR; high-speed running; ^a^ denotes difference from MD-4; ^b^ denotes difference from MD-2; ^c^ denotes difference from MD; all *p* ≤ 0.05; * means a very large effect size for all differences (>2.0).

**Table 2 healthcare-10-00695-t002:** External Intensity by accelerometery-based variables during training and matches for squad average (mean ± SD).

MD	Player Load (AU)	ACC1	ACC2	ACC3	ACC4	DEC1	DEC2	DEC3	DEC4
MD-5	284.4 ± 11.7 ^a,b,^*	138.6 ± 7.6 ^b,c,^*	80.9 ± 4.3 ^b,c,^*	29.9 ± 2.7 ^a,b,^*	9.6 ± 1.4	126.3 ± 6.8 ^b,c,^*	77.8 ± 4.6 ^b,c,^*	28.0 ± 2.4 ^b,c,^*	11.8 ± 1.6 ^c,^*
MD-4	263.4 ± 10.5 ^b,c,^*	134.8 ± 6.7 ^b,c,^*	79.8 ± 3.5 ^b,c,^*	26.7 ± 2.5 ^b,c,^*	7.5 ± 1.2	121.9 ± 5.6 ^b,c,^*	77.6 ± 3.6 ^b,c,^*	27.9 ± 2.2 ^b,c,^*	11.2 ± 1.4 ^c,^*
MD-2	222.6 ± 7.2 ^c,^*	100.7 ± 4.1 ^c,^*	52.9 ± 1.7 ^c,^*	21.3 ± 1.3 ^c,^*	9.8 ± 1.0	89.6 ± 4.1 ^c,^*	56.0 ± 2.3 ^c,^*	21.2 ± 1.3 ^c,^*	9.7 ± 1.1 ^c,^*
MD	324.4 ± 14.0	177.3 ± 8.2	106.8 ± 6.4	34.9 ± 3.4	10.3 ± 1.8	169.0 ± 4.8	98.8 ± 5.9	39.0 ± 4.1	18.9 ± 2.9

MD, match-day; MD- = matchday minus (5, 4, 2); AU, Arbitrary Units; ACC, acceleration; DEC, deceleration. Both ACC and DEC were measured in number (counts); ^a^ denotes difference from MD-4; ^b^ denotes difference from MD-2; ^c^ denotes difference from MD; all *p* ≤ 0.05; * means a very large effect size for all differences (>2.0).

**Table 3 healthcare-10-00695-t003:** Internal Intensity and Wellness Profile during training and matches for squad average (mean ± SD).

MD	RPE (AU)	s-RPE (AU)	Fatigue (AU)	Stress (AU)	DOMS (AU)	Sleep Quality (AU)	HI (AU)
MD-5	5.9 ± 0.3 ^b,c,^*	508.3 ± 29.0 ^b,#^	3.3 ± 0.2	3.6 ± 0.4	2.9 ± 0.3	3.6 ± 0.3	13.4 ± 0.8
MD-4	5.4 ± 0.2 ^b,c,^*	473.7 ± 20.5 ^b,c,^*	3.4 ± 0.2	3.1 ± 0.3	3.0 ± 0.3	3.4 ± 0.2	13.0 ± 0.7
MD-2	4.4 ± 0.3 ^c,^*	353.5 ± 21.9 ^c,^*	3.3 ± 0.2	3.2 ± 0.4	2.7 ± 0.2	3.3 ± 0.2	12.5 ± 0.7
MD	7.9 ± 0.3	604.7 ± 36.5	3.1 ± 0.1	3.0 ± 0.2	3.0 ± 0.2	3.3 ± 0.3	11.6 ± 0.8

MD, match-day; MD-, matchday minus (5, 4, 2); AU, Arbitrary Units; RPE, rated perceived exertion; s-RPE, session-RPE; DOMS, delayed onset muscle soreness; HI, Total Hooper Index; ^a^ denotes difference from MD-4. ^b^ denotes difference from MD-2. ^c^ denotes difference from MD; all *p* ≤ 0.05; ^#^ means a large effect size (>1.2–2.0); * means a very large effect size for all differences (>2.0).

## Data Availability

The data presented in this study are available on request from the corresponding author.

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
