# Peer review of "In-Season Microcycle Quantification of Professional Women Soccer Players—External, Internal and Wellness Measures"

_healthcare, 2022, doi:10.3390/healthcare10040695_

Round 1

Reviewer 1 Report

  1. "In-season Microcycle Quantification of Professional Women Soccer Players – External, Internal and Wellness Measures" is a topic of critical significance to the science of training in sport.
  2. The background, methodology, results and discussion have the potential to be impactful to the discipline of sport and training science.
  3. Nevertheless, there is serious need for re-writing the manuscript, English-wise, so that the readers can get the gist of what the research is trying to convey. This shortcoming in the English expression is evident from the first paragraph of the introduction. The sentence does not convey accurately what it is supposed to say: "Women's soccer is growing in a number of participants [1–3] which is conducting to the progressive raising of publications related to the topic (in the PubMed, a simple search by “women” AND “soccer” revealed an increment of publications from 75 in 2019 to 125 in 2021). This reveals the concerns of community science to offer more evidence to practical community."
  4. I suggest to the authors to find somebody proficient in English who can go through the manuscript and re-write with clarity the problematic areas in the introduction, materials and methods, and the concluding statement. 

Reviewer 2 Report

Line 27: Please rephrase to “Although data currently exists pertaining to the intensity…”

Line 65-71: Can you provide a little more context for this? What part of the season or training cycle were the heavy weeks in comparison to the lighter weeks? Did these include matches?

Line 78-83: Can you be more specific here in order to justify your research questions? Which particular reports and what interesting findings regarding the associations between physiological and physical demands? Can you highlight some of the potential limitations of studies included in the previous systematic review that are addressed with the current investigation? Why can it be expected to observe intra-week variations in wellness?

Line 96: I believe this hypothesis can be expressed in a more elegant manner. Perhaps with a correlation between Training Intensity and Hooper Index values?

Line 121: Were the number of players for each position (outside of goalkeeper) randomly determined? More explanation of sample size was distributed amongst the positions would be helpful.

Line 129: I believe this reference, [24], needs to be moved to a different location

Line 169: I may have missed the explanation, but is MD-3 and MD-1 missing?

Line 203: Should restate research question and hypothesis and whether the hypothesis was rejected or accepted.

Line 221: Maybe hopefully to other readers to explain pyramidal weekly microcyle shape for training intensity distribution.

Line 235: I would suggest staying away from confirmed and change to in agreement with…

Reviewer 3 Report

Dear authors,

First, I would like to congratulate you on conducting this research which is a huge of value considering that how fast women's soccer is growing. I have some comments  which I hope that they will help to improve the quality of your manuscript.

  1. It would be worthwhile touching on menstrual cycle in women and how that might affect RPE during training and matches.  
  2. Any serious injury to report? Either way, please elaborate.
  3. "Training and competition processes trigger the formation of stress markers such as 204 augmented levels of stress and reduced immunoglobulin A in team-sport athletes [31].." Please remove this line as it's irrelevant to your measured data. 

  4. Any control for their dietary intake and sleep? if not, please acknowledge this as a limitation.
  5. What are your practical applications? I think there should be a paragraph outlining how these results can be applied in practical setting to ensure health and safety of women soccer players. 
